# Behavioral and environmental determinants of acute diarrhea among under-five children from public health facilities of Siyadebirena Wayu district, north Shoa zone, Amhara regional state, Ethiopia: Unmatched case-control study

Behailu Tariku Derseh [1]☯*, Natnael Mulushewa Tafese[2]☯, Hazaratali Panari[3]‡, Awraris Hailu Bilchut[1]‡, Abel Fekadu Dadi[4,5]‡

1 Department of Public Health, Asrat Woldeyes Health Sciences Campus, Debre Berhan University, Debre Berhan, Amhara, Ethiopia, 2 Waghimira Health Department, Amhara Region Health Bureau, Health System Strengthening Special Support Directorate, Federal Ministry of Health, Addis Ababa, Ethiopia, 3 Department of Nursing, Asrat Woldeyes Health Sciences Campus, Debre Berhan University, Debre Berhan, Amhara, Ethiopia, 4 Menzies School of Health Research, Charles Darwin University, Darwin, Australia, 5 Department of Epidemiology and Biostatistics, College of Medicine and Health Sciences, University of Gondar, Gondar, Amhara, Ethiopia

☯ These authors contributed equally to this work.
‡ These authors also contributed equally to this work.
* minastariku@gmail.com

## Abstract

### Background

Acute diarrhea is a major public health problem in the world. Next to pneumonia, it is the leading cause of death in children under five years old. Globally, even though childhood diarrhea disease kills millions, the interaction of socio-demographic, behavioral, and environmental factors of acute diarrhea in children aged 6–59 months is not investigated yet in the current study area.

### Objective

To determine behavioral and environmental predictors of acute diarrhea among under-five children from public health facilities of Siyadebirena Wayu district, North Shoa, Amhara Regional State, Ethiopia, 2019.

### Methods

A facility-based unmatched case-control study was conducted from March 12, 2019, to May 12, 2019. A total of 315 under-five children were included in the study (105 cases and 210 controls). A systematic random sampling technique was used to select study participants. Data were collected by a structured questionnaire and analyzed by using SPSS. To analyze the data, bivariable and multivariable logistic regression analysis was used.

**Data Availability Statement:** All relevant data are presented within the paper and its Supporting Information files.

**Funding:** The authors received no specific funding for this original research.

**Competing interests:** The authors have declared that no competing interests exist.

**Abbreviations:** CDC, Communicable Disease Control; CSA, Central Statistical Agency; DBU, Debre Berhan University; EDHS, Ethiopian Demographic Health Survey; IMNCI, Integrated Management of Neonatal and Childhood Illness; IPD, In-patient Department; OPD, Out Patient Department; OR, Odds Ratio; SPSS, Statistical Package for Social Sciences; USAID, United States Aid for International Development; WHO, World Health Organization.

## Results

The study showed that average family monthly income of 12–23 USD (AOR = 6. 22; 95% CI: 1.30, 29.64), hand washing practice of mothers/ care givers with water only (AOR = 3.75; 95% CI: 1.16, 12.13), improper disposal of infant feces (AOR = 11.01; 95% CI: 3.37, 35.96), not treating drinking water at home (AOR = 9.36; 95% CI: 2.73, 32.08), children consuming left-over food stored at room temperature (AOR = 5.52; 95% CI: 1.60, 19.03) and poor knowledge of the respondents about the risk factors for diarrhea were the determinants that significantly associated with acute childhood diarrhea.

## Conclusion

The potential predictors of childhood diarrhea morbidity were improper hand-washing practice, not treating drinking water at home, unsafe disposal of children's feces, children consuming left-over food stored at room temperature, and having poor knowledge about the major risk factors for diarrhea. Thus, awareness of the community on hygiene and sanitation focusing on proper handling of human excreta, safe water handling, proper hand washing practice, and proper management of leftover food should be enhanced to prevent children from acute diarrhea diseases.

## Introduction

The World Health Organization defines diarrhea as the passage of three or more loose or liquid stools per day [1]. It is caused by bacterial, viral, and parasitic organisms and is usually causes gastrointestinal infection. It can be transmitted through the fecal-oral route and is spread through contaminated food and drinking water or from person to person as a result of poor hygiene and sanitation practice [2]. Whereas acute diarrhea, a major public health problem, is characterized by abrupt onset of frequent, watery, loose of stools with blood or without blood, and mucus in feces lasting less than two weeks. Usually, acute watery diarrhea episodes subside within 72 hours of onset. It may be accompanied by flatulence, malaise, and abdominal pain. Nausea, vomiting may occur and also fever may be present [3].

Next to pneumonia, it is the leading cause of death in children under-five years old. Globally, there are nearly 1.7 billion cases of childhood diarrheal disease every year [2]. In developing countries, acute gastroenteritis is one of the most common causes of consultation in the emergency room and admission among the pediatric age group [4]. Morbidity in young children is a serious problem because early childhood is a critical period in terms of development. Physical growth and cognitive pathways during this period are faster than during any other time. Disruption of these processes by acute diarrhea in the short term can lead to mortality and have long term consequences [5].

Annually, an estimated 1.3 million under-five deaths are attributed to diarrheal diseases, and most affecting children in resource-limited countries [5]. Young children in Africa and South-east Asia are most vulnerable with the incidence of severe gastroenteritis being highest in the first 2 years of life. Morbidity due to diarrhea is further concentrated in marginalized communities within resource-limited countries. Despite improvements in the standard of living, advances in sanitation, water treatment, and food safety awareness, diarrheal disease still accounts for significant economic and societal losses [5]. In Ethiopia, diarrheal diseases are the major contributors to under-five mortality. According to the 2016 Ethiopia Demographic and

Health Survey report,12% of under-five children had a diarrheal episode and among sick under 5 children, about 56 percent for whom advice or treatment was not sought in the 2 weeks before the survey [6]. Studies conducted in different parts revealed that, diarrhea is one of the common causes of under-five mortality; the prevalence's ranges from 8% to 32% that is in Mecha district a community based, cross sectional study (8%) [7], Farta district 17%, Dejen district, a community based, cross-sectional study (24%) [8], Jabithennan District, a community based, cross-sectional study (25%) [9], a systematic review and meta-analysis conducted based on 31 studies revealed that 27% in Afar region, 26% in Dire-Dawa, and 24% in Addis Ababa [10].

Socioeconomic status (SES) affects health care quality and education. Moreover, SES affects the diet, housing conditions, and increases the likeliness of acquiring infectious diseases. Children in households with lower socioeconomic status receive oral rehydration therapy less often than children in households with higher socioeconomic status [1, 7, 11]. The same is true in Siyadebirena Wayu district, North Shoa, Amhara Regional States Ethiopia in which acute diarrhea is the common problem of children aged 6–59 months. As of the 2017 or 2018 annual performance report of the health office, it was the top leading causes of under-five morbidity in the district. However, there was no scientific evidence of causative factors [12]. Thus, this study was aimed to assess behavioral and environmental determinants of acute diarrhea among under-five children from public health facilities at Siyadebirena Wayu district, North Shoa Zone, Amhara, Ethiopia.

## Methodology

### Study area

The study was conducted in Siyadebirena Wayu district public health institutions. Siyadebirena Wayu district is one of the 24 districts in North Shoa Zone of the Amhara Region. The total population of the district in the year 2019 was estimated to be 73,471 in which 9,948 are under-five children. The district consists of one urban Kebele, the smallest administrative unit, (Deneba town) and 13 rural Kebeles. Deneba, the capital town of the district, is located at about 129 Km from Addis Ababa, 560 km from Bahir-Dar, and 47 km from Debre Berhan. The district has 14 health posts, 3 governmental health centers, 1 primary hospital, 4 private clinics, and 1 private pharmacy. Rivers and hand-dug wells are the main sources of water for rural areas. The majority of the town population obtains piped water from deep wells. The most common health problems of children in Siyadebirena Wayu district are communicable diseases like pneumonia and diarrhea [12].

### Study design and study period

An institution-based unmatched case-control study was conducted from March 12[th] to May 12[th], 2019 in Siyadebirena Wayu district, North Shoa, Amhara regional state, Ethiopia. Two health facilities, namely, Deneba health center and Deneba hospital were included in the study.

### Selection of cases and controls

The cases were selected children with acute diarrhea in the pediatric outpatient department (OPD) at the health center or OPD and pediatric ward in case of Deneba primary hospital coming for treatment from March 12[th] to May 12[th], 2019. During the selection of cases, acute diarrhea was identified using WHO signs and symptoms for diarrhea [2]. However, since there is no specific sampling frame for the selection of cases, systematic random sampling technique was used considering daily fluctuations and the average number of cases in the

previous two months. Thus, based on the previous 2 months' performance of health facilities, and registered at under-five Integrated Management of Neonatal and Childhood Illness (IMNCI) registration books, the expected number of under-five children with acute diarrhea was taken. Then, the sample size for cases was proportionally allocated for each health institution. Finally, during the data collection period, the data collectors gathered information from the cases in each public health institution until allocated sample size was achieved. Similarly, the controls were recruited from the same OPD/ ward of health center and hospital during the same period. Individual selection was carried out one case at a time by selecting controls from the immediate public health institutions.

**Inclusion criteria for cases and controls.** Children aged 6–59 months, who had three loose and watery stools within 24hrs period for the last consecutive 3 days with the determination of physician as acute diarrhea, were enrolled as cases [13]. Conversely, children aged 6–59 months without acute diarrhea were enrolled as controls. However, children aged 6–59 months whose mothers or care-takers could not respond to the questionnaire due to health-related problems, and children with chronic diseases were excluded from the study for either case.

## Sample size determination

The sample size was determined using EPI Info Version 7 statistical software by considering the following assumptions: 95% confidence level (1.96), 80% power, P1 = 44.5%, proportion of diarrheic children whose family dispose of their infant feces in the latrine, P2 = 27.1%, proportion of children non-diarrheic children whose family dispose of their infant feces in the latrine as main predictors of the outcome variable from studies conducted in the Chire district, Ethiopia[14]. The proportion of case and control was assumed to be 1:2. Therefore, considering a 10% of non-response rate, the final sample size calculated was 315 (105 cases and 210 controls).

## Sampling techniques

Siyadebirena Wayu district has 3 health centers and 1 primary hospital. Deneba Health Center was selected randomly and Deneba primary hospital was included in this study due to accessibility and feasibility reasons. A systematic random sampling technique was used to select 315 study participants. Cases were proportionally allocated to the health centers and primary hospital-based on the previous 2 months' experience. By considering the last two months' performances of under-five children who visited health institutions cases and controls were selected by a systematic random sampling technique. Hence, K (sampling interval) was 2 for cases and 3 for controls. The first case and control to be included in the sample were chosen randomly, then every 2nd for cases and every 3rd for controls were taken until the sample size was reached (Fig 1).

## Variables of the study

**Dependent variable.** Acute diarrhea.

**Independent variables.** These variables were sub-divided into three divisions. **Socio-demographic status:** include family economic status, place of residence, household size, maternal age, education, ethnicity, number of children, occupation, marital status, religion, child age, birth order, and sex of the child. Environmental sanitation: include type of water source, distance to the water source, amount of daily water consumption, availability of latrine, number of rooms, livestock in house, refuse disposal, housing conditions. Behavioral factors: include method of water drawing and storage, feeding practices, action for diarrhea, duration of breastfeeding, time of introducing supplementary feeding, knowledge about major risk

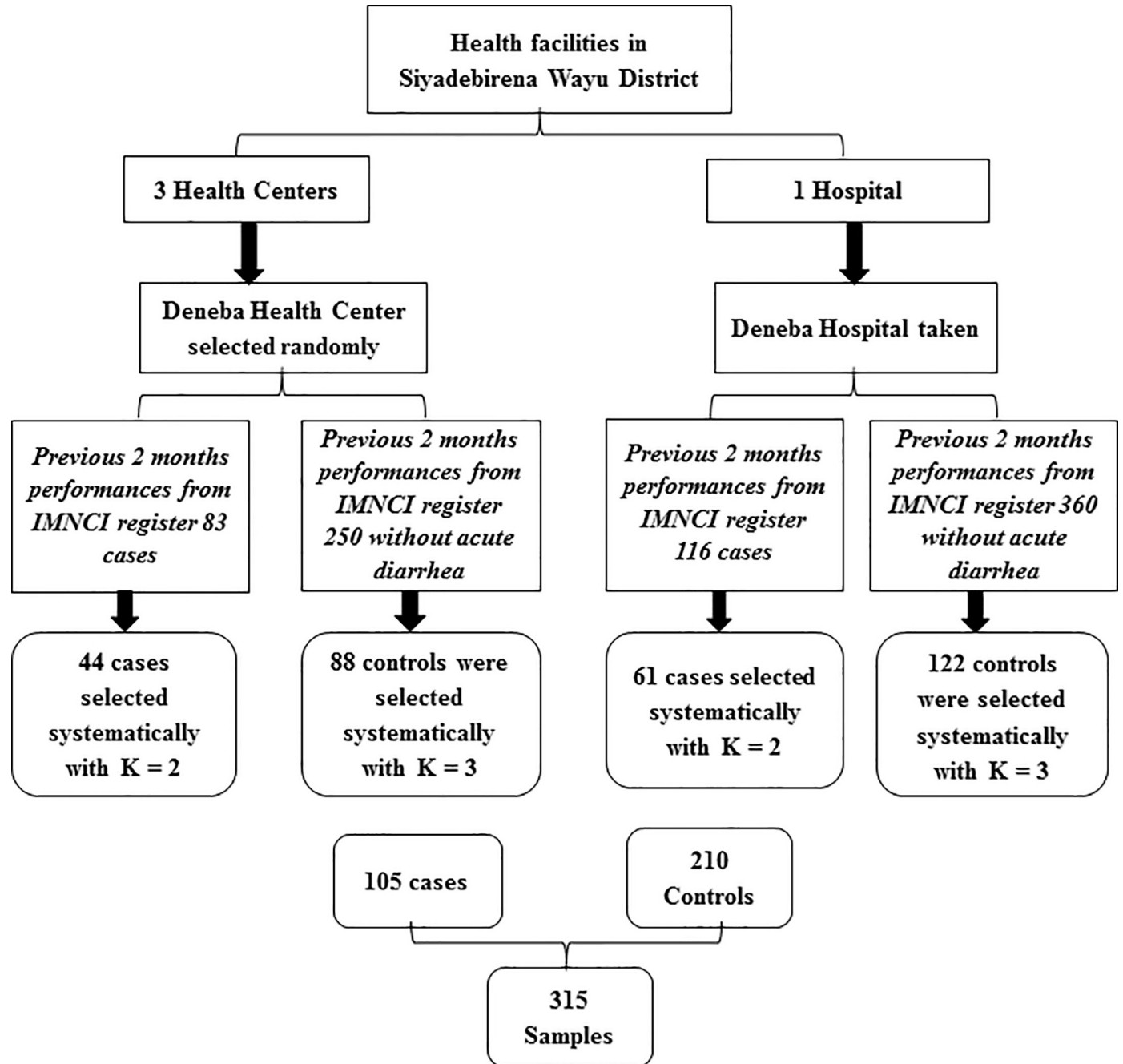

**Fig 1. Sampling procedures for cases and controls in Siyadebirena Wayu district, North Shoa Zone, Amhara region, Ethiopia, 2019.**

factors, management of leftover food, recent maternal history of diarrhea, hand washing practice, hand washing material, Rota virus vaccination, measles vaccination, vitamin A supplementation.

### Data collection tools and methods

Data were collected using a pretested structured interviewer-administered questionnaire prepared by reviewing previous studies and other materials. The questionnaire contains three

sections such as socio-demographic, environmental sanitation, and child care behavioral related variables (S1 Annex). Data was collected by 5 nurses who works at the under-five OPD/inpatient pediatrics ward of the health center and hospital.

## Data quality management

The questionnaire was developed by reviewing different works of literature. To keep its consistency, a questionnaire first prepared in English was translated to Amharic and then back to English. Objective-based, logically sequenced, free of scientific terms, and non-leading structured questionnaire was prepared. A pretest was undertaken on the questionnaire before the actual data collection started. It was undertaken on 30 individuals and an amendment was taken on the questionnaire. The finding of the pretest was discussed among data collectors, supervisors, and researchers to ensure a better understanding of tools and procedures so that it was modified accordingly to final. Moreover, data collectors and supervisors were provided with intensive training for two days on the objective of the study, contents of the questionnaires, and how to maintain confidentiality and privacy of the study subjects. The assigned supervisors made a day to day on-site supervision during the whole period of data collection and checked the collected data for completeness, clarity, and consistency on a daily basis.

The collected information was rechecked for its completeness and consistency before entering the data into a computer. Ten percent of the data were re-entered to see its validation. At the end of data entry, data cleaning was made. Frequencies, cross-tabulations, sorting, and filters were used to check missed values and variables. Errors identified were corrected after revising the original questionnaire.

## Data processing and analysis

The data were entered into EPI-data version 3.1 for windows and exported to SPSS 20 for windows for analysis. The first step before analysis was data exploration to visualize the general feature of the data to be analyzed. After exploration, bi-variate analysis and multi-variable analysis were performed step by step.

The bi-variate analysis using cross-tabulation and bi variable logistic regression was done. Bi-variate analysis using cross-tabulation was done to determine the distribution of study subjects by independent variables of interest. Bivariate logistic regression technique was done to see the crude association between the independent variables and the dependent variable.

The final step of the analysis was multivariable analysis using a hierarchical logistic regression technique to assess the relative effect of the explanatory variables on the outcome variable. Independent variables which result in a p-value less than 0.20 [15] in an unadjusted model are candidates to be considered for the final multivariable model. Multivariable logistic regression was fitted to obtain adjusted odds ratios (AOR) after controlling the confounding effects of different variables and to determine factors associated with the outcome variable. Significance level at p-value less than 0.05 with a 95% confidence interval (CI) was taken to decide that there is a significant association between outcome and explanatory variables. Hosmer-Lemeshow goodness of fit test was performed to check the adequacy of the final model (Table 1).

## Ethical approval and consent to take part

The ethical clearance was obtained from the Ethical Review Committee (ERC) of Debre Berhan University; Health Sciences College (Protocol no. 19/19/SPH; date: 19/03/2019). Permission was obtained from Siyadebirena Wayu district administration, district health office, Health Centers, and Hospital. The purpose of the study was explained to the mothers/caregivers, and verbal consent was obtained. The confidentiality of information was maintained

**Table 1. Model describing the hierarchical logistic regression analysis on the determinants of acute diarrheal disease among children aged 6–59 months in public health facility in Siyadebirena Wayu district, Ethiopia, 2019.**

| Model | Block $x^2$(P-value) | Hosmer- Lemeshow goodness of-fit-test |
|---|---|---|
| | | (p-value) |
| Model 1 | 61.05 (p < 0.001) | 0.263 |
| Model 2 | 149.64 (p < 0.001) | 0.792 |
| Model 3 | 233.17(p < 0.001) | 0.986 |

Model 1: Examined the joint effects of socio-demographic factors

Model 2: Used socio-demographic factors with p-value < 0.20 in Model 1, and environmental factors

Model 3: Built on variables with p-value < 0.20 in Model 2 and included behavioral factors.

during the interview process by avoiding unique personal identification information. Moreover, information about the purpose of the study, the rules, the risks and benefits of this research was provided for all study participants.

## Results

### Socio-demographic characteristics of the respondents

A total of 309 under-five children's mothers/care givers were included in this study giving a response rate of 98%. Among these, 99 (96.1%) of cases and 200 (97.1%) of controls were biological mothers. Forty-nine (49/103, 47.6%) of mothers in cases and 108 (52.4%) of mothers from controls were found in the age group of 25–34 years. The mean (± SD) age of mothers was 32.31 (±5.6) years for cases and 32.18 (±5.6) for controls (S1 Table).

### Environmental characteristics of the respondents

Out of the respondents, 90 (87.4%) of cases and 202 (98.1%) of controls have been using improved water sources. Sixty of cases and one hundred nineteen controls reported that they spent less than 15 minutes (round trip) to fetch drinking water. But 86(83.5%) of cases and 56 (27.2%) of controls, did not treat their drinking water at home. In addition, house-holds of 25 (24.3%) of cases and 40 (19.4%) of controls did not have latrines."S2 Table" illustrates the environmental characteristics in detail.

### Behavioral characteristics of study participants

In this study, 82(79.6%) of cases and 120(58.3%) of controls mothers/caregivers had poor hand washing practices. Out of 280 mothers/caregivers who practices hand washing 54(68.4%) of cases and 37(18.4%) of controls were washing their hands with water only and the rest used soap and water to wash their hands. With regard to vaccination status, 34 (33%) of cases and 28(13.6%) of controls were not vaccinated for measles, 27(26.2%) of cases 15(7.3%) of controls did not vaccinate for Rota Virus, and 41(39.8%) of cases and 33(16%) of controls did not receive vitamin-A supplementation. Moreover, 90(87.4%) of cases and 77 (37.4%) of controls mothers/care-takers have poor knowledge about major risk factors of acute diarrhea. The details of behavioral characters are explained in "S3 Table".

### Determinants of acute diarrhea among under-five children

After controlling the confounding effects of independent variables, the following variables were statistically significant in a multivariable analysis at a 5% significance level. These predictors were family monthly income (AOR = 6.22; 95% CI: 1.30, 29.64), hand washing without

soap (AOR = 3.75, 95% CI: 1.16–12.13), families who did not treat their drinking water at home (AOR = 9.36; 95% CI: 2.73, 32.08), families who dispose infant feces outside the latrine (AOR = 11.01; 95% CI: 3.37, 35.96), mothers/caregivers who had poor knowledge about the major risk factors of acute diarrhea (AOR = 15.3; 95% CI: 4.18, 55.88), and families who consume leftover food at room temperature (AOR = 5.52; 95% CI: 1.60, 19.03) (Table 2).

## Discussion

Childhood diarrhea diseases have been hypothesized by different studies to be associated with socio-demographic, environmental, and behavioral factors. This study tried to assess potential amenable factors of acute diarrhea among children aged 6–59 months at the health facility level.

The average monthly income of the family was one of the predictors of childhood acute diarrhea. Children whose family average monthly income between 12–23 USD was about 6 times more likely to contract acute diarrhea than children whose income was more than 24 USD. This finding was similar to studies conducted at Gaza strip where the richer children's family the lesser to develop diarrhea [16]. These studies state that children living in poor households have higher rates of infection with acute diarrhea than their wealthier counterparts. The reasons probably could be because of inadequate access to sanitary facilities, unsanitary environments in the home, and poor hygienic practice of children's parents. Moreover, children from these household could not afford clean and safe food. At the same time, rich families may have greater opportunity to use soap for handwashing and aqua-guard at their houses to protect microbial contamination in water, and they may construct toilets. However, the monthly income of the family was not significant in other studies [11, 17]. The possible explanation for this difference might be explained by socio-demographic variations between study participants.

The current study showed that children from mothers who washed their hands without soap were nearly 4 times more likely to develop acute diarrhea than those children whose mothers washed their hands with water, and soap. In a parallel way, a study done on under-five diarrhea in Jabithennan district reported that being from mothers of poor handwashing practice was significantly associated with childhood diarrhea disease (AOR = 5.53; 95% CI: 2.19, 13.99) [9]. Similarly, studies conducted at Dejen, Northwest Ethiopia (AOR = 1.61; 95% CI: 1.04, 2.84) [8], Hadaleala, Afar Region, Northeast Ethiopia (AOR = 24.94; 95% CI: 6.68, 93.12) [18], and Farta district, North West Ethiopia (AOR = 1.59; 95% CI: 1.11, 2.27) [19] found that mothers handwashing practices without soap were associated with increased risk of acute diarrhea in their children. A study was done in the Philippines strengthen the idea; handwashing with soap is most effective in reducing acute gastroenteritis by 42–47% [4]. A study conducted at Keresa district [20] supported this evidence too.

The quality of the drinking water supply is another predictor of childhood diarrhea. It was found that children whose families did not treat drinking water at their homes were 9 times more likely to develop acute diarrhea compared with those families who treated water for drinking purpose. This finding is in agreement with study from Pawi Hospital (AOR = 2.46; 95% CI: 1.32, 4.57) [17], Derashe district, southern Ethiopia (AOR = 2.25; 95% CI: 1.43, 3.56) [21], Wolaita Soddo town (AOR = 2.34; 95% CI: 1.33, 4.14) [22]. However, it contradicts with previous studies in Northern Gondar and Yaya Gulele in which household water treatment was not significant for acute diarrhea [23, 24]. This can be justified by the fact that collected water is liable for contamination during collection, transportation, and storage which may, in turn, increase the risk of diarrheal diseases. In addition, the discrepancy might be explained by design difference since other studies, unlike the present study, used cross-sectional study design.

**Table 2. Factors associated with acute diarrhea among children aged 6–59 months who visited selected public health facilities in Siyadebirena Wayu District, North Shoa Zone, Amhara region, Ethiopia, 2019.**

| Variables | Frequency | | COR (95% CI) | p-value | AOR (95% CI) |
|---|---|---|---|---|---|
| | Cases | Controls | | | |
| Average family income | | | | | |
| <12 USD | 46 | 62 | 3.68 (2.07, 6.54) | 0.065 | 3.42 (0.92, 12.64) |
| 13–23 USD | 32 | 20 | 7.94 (3.92, 16.06) | 0.022 | 6.22 (1.30, 29.64)* |
| ≥ 24 USD | 25 | 124 | 1 | | 1 |
| Separately prepare food | | | | | |
| Yes | 86 | 193 | 1 | | 1 |
| No | 17 | 13 | 2.90 (1.38, 6.10) | 0.150 | 3.72 (0.62, 22.33) |
| Consume leftover food | | | | | |
| No | 45 | 157 | 1 | | 1 |
| Yes | 58 | 49 | 4.13 (2.49, 6.83) | 0.002 | 5.52 (1.60, 19.03)* |
| Water source | | | | | |
| Protected | 90 | 202 | 7.29 (2.31, 22.98) | 0.150 | 2.85 (0.26, 31.12) |
| Unprotected | 13 | 4 | | | |
| Treating water at home | | | | | |
| Yes | 17 | 150 | 1 | | 1 |
| No | 86 | 56 | 13.55 (7.41, 24.79) | 0.001 | 9.36 (2.73, 32.08)* |
| Hand washing | | | | | |
| Without soap | 54 | 37 | 9.57 (5.29, 17.33) | 0.023 | 3.75 (1.16, 12.13)* |
| With soap and water | 25 | 164 | 1 | | 1 |
| Disposing of child feces | | | | | |
| Inside the latrine | 17 | 53 | 1 | | 1 |
| Outside the latrine | 86 | 153 | 14.64 (7.96, 26.79) | 0.001 | 11.01 (3.37, 35.96)* |
| Measles vaccination | | | | | |
| Yes | 69 | 178 | 1.00 | | 1.00 |
| No | 34 | 28 | 3.13 (1.77, 5.55) | 0.849 | 1.24 (0.13, 11.36) |
| Rotavirus vaccine | | | | | |
| Yes | 76 | 191 | 1 | | 1 |
| No | 27 | 15 | 4.52 (2.52, 8.97) | 0.804 | 0.79 (0.12, 4.94) |
| Vit A supplementation | | | | | |
| Yes | 41 | 33 | 1 | | 1 |
| No | 62 | 177 | 3.47 (2.01, 5.96) | 0.272 | 3.54 (0.47, 26.79) |
| Mothers recent history of diarrhea | | | | | |
| Yes | 29 | 25 | 2.84 (1.56, 5.17) | 0.325 | 0.49 (0.12, 12.2.02) |
| No | 74 | 181 | 1 | | 1 |
| Hand washing practices | | | | | |
| Poor | 82 | 120 | 2.80 (1.61, 4.87) | 0.566 | 1.49 (0.36, 5.84) |
| Good | 21 | 86 | 1 | | 1 |
| Knowledge on major risks of diarrhea | | | | | |
| Poor | 90 | 77 | 11.60 (6.08, 22.14) | 0.001 | 15.30 (4.18, 55.88)* |
| Good | 13 | 129 | 1 | | 1 |
| Knowledge to action taken | | | | | |
| Poor | 69 | 36 | 9.58 (5.55, 16.54) | 0.001 | 14.09 (4.08, 48.62)* |
| Good | 34 | 170 | 1 | | 1 |

Note: COR = Crude Odds Ratio, AOR = Adjusted Odds Ratio

* = Significant variables at P-value less than 0.05.

The proper disposal of children's feces is extremely important in preventing the spread of diarrhea disease. Contact with human feces directly, or indirectly by animal, can lead to diarrhea diseases. In this study, families who dispose infant feces outside the latrine were 11 times more likely to develop acute diarrhea compared with children whose families disposed infant feces inside the latrine. Similarly, study conducted in Dejen district (AOR = 1.53; 95% CI: 1.05, 2.24) [8], Chire district (AOR = 3.69; 95% CI: 1.13, 5.93) [14], Pawi Hospital (AOR = 2.72; 95% CI: 1.54, 4.81) [17], West Gojjam (AOR = 1.90; 95% CI: 1.12, 3.22) [25], and Benishangul Gumuz Regional State (AOR = 0.49; 95% CI: 0.34, 0.78) [26] showed the association of diarrhea diseases and improper disposal of human feces. In a similar fashion, studies done in Nigeria and Indonesia also supported that safe and proper disposal of children's feces is highly important in preventing the spread of disease, as direct contact with human feces can cause diarrhea and/or other related infectious diseases [27, 28]. This implies that the safe disposal of feces can inhibit the direct contamination of farmed crops, indirect contamination of water supplies, a breeding place for flies, and a source of the fecal pathogens that flies can spread. However, some studies reported that proper disposal of child feces is not significantly associated with diarrhea diseases[29, 30]. The difference might be attributed to methodological variation and socio-demographic variations of study participants.

Since diarrhea disease transmits from person-to-person, leftover food can act as a mechanism for indirect transmission of diseases. Our study showed that children who consumed left-over food stored at room temperature were 5.52 times more likely to have diarrhea compared with children who did not consume left-over food. This pattern was consistent with studies conducted in Derashe district in which children who consumed left-over food stored at room temperature were more likely to have diarrhea compared with children who did not consume (AOR = 1.65; 95% CI: 1.01, 2.71) [21]. This idea was also supported by a similar study conducted in Hadaleala district, Afar Region, Northeast Ethiopia, which strengthens that diarrheal disease was highly prevalent among children who didn't eat foods immediately after cooking (AOR = 3.74; 95% CI: 1.48, 9.45) [18].

Moreover, as knowledge is the foundation for committing healthy behavior, children's mothers/ care givers with poor knowledge on the major risks of acute diarrhea were 15 times more likely to develop acute diarrhea than children whose families had good knowledge. Similarly, a study conducted in Chire district, Ethiopia reported that the maternal or caretakers' knowledge has a significant preventive and control effect on diarrhea disease (AOR = 4.00; 95% CI: 2.52, 6.35) [14]. This could be explained as the more the respondents are knowledgeable about the mechanism of disease transmission, the more they practice the different preventive measures [31]. This study also showed that mothers/ caretakers' knowledge on appropriate actions taken to their children when they had acute diarrhea was independently associated. Thus, the lesser appropriate action taken corresponds to 14 times higher risk of diarrhea among children with poor knowledge about the actions taken to the child. This result implies that it is possible to reduce acute diarrhea by increasing awareness of mothers about actions taken to acute diarrhea. On the contrary, a study done in Surakarta reported that poor knowledge of mothers about healthy life is a risk factor that causes diarrhea in infants (AOR = 2.30; 95% CI: 3.46, 1.14) [31]. Similarly, developing diarrhea was higher among children whose mothers/caretakers had better knowledge about the causes of diarrhea and had handwashing practice (AOR = 2.46, 95% CI: 1.07, 5.63) [22]. However, a study done in Arba Minch Zuria district stated that knowledge is not a significant predictor [32]. This disagreement might be attributed to methodological differences as the study done in Arba Minch district used a cross-sectional study design.

In summary, the findings of this study have a paramount implication for the control of diarrhea morbidity among under-five children. It would provide helpful insights for stakeholders

and program implementer working on the potential risk factors of acute diarrhea to take priority interventions in order to prevent and control the disease. One of the strengths of this study was its design being a case-control, and cases and controls confirmation was done by a physician provides strong evidence for the association. However, this study did not identify the causative agents and the pathogens involved in different factors. For example what type of microbes were involved during improper handwashing, untreated drinking water, and consumption of leftover food were not explained. Other limitations can be, the results might have been biased because of potential recall bias; however, this was minimized by using reported-incident cases within two weeks. Some behavioral practices including handwashing practices used in the analysis were self-reported by the respondents; self- reported data have been found to introduce inaccuracy and bias into estimates of behavior. Moreover, the wider confidence interval as a result of smaller frequency in some categories of predictor variables may reduce the precision of the measure of association. Therefore, the use of this study findings should be considered as having these inherent limitations.

## Conclusion and recommendations

In conclusion, the potential determinant factors for childhood diarrhea morbidity were improper hand washing, not treating drinking water at home, unsafe disposal of child feces, consumption of leftover food at room temperature, mothers' poor knowledge about the major risk factors for diarrhea, mothers' poor knowledge on action to be taken after children develop acute diarrhea, and low monthly income of the family. Therefore, health workers are highly recommended to take actions on the contextual behavioral factors of acute diarrhea and should provide regular health education on the prevention and control of childhood acute diarrhea. District health office should plan strategies for the distribution of disinfectant to treat drinking water at home level and to reach the rural community where the risk of water-borne diseases is high. Furthermore, effective educational programs that emphasize hand washing practice with soap at critical period, safe disposal of human excreta, community lead total sanitation, and promote eating foods immediately after cooking, avoid eating leftover foods should be strengthened in collaboration with health extension workers' who integrate water hygiene and environmental sanitation while conducting a home visit. Lastly, other researchers are advised to further identify the causative agents and the pathogens involved in different risk factors of acute diarrhea.

## Supporting information

**S1 Table. Distribution of study participants by socio-demographic characteristics.**
(PDF)

**S2 Table. Distribution of environmental conditions of study participants.**
(PDF)

**S3 Table. Behavioral factors of study participants in relation to acute diarrhea.**
(PDF)

**S1 Annex. English and Amharic version questionnaires.**
(PDF)

**S1 Data. All data in this article is available in S1 Data.**
(SAV)

## Acknowledgments

We would like to thank the Department of Public Health, College of Health Sciences, Debre Berhan University for their facilitation and guidance to undertake this work. Besides, we are very glad to forward our special thanks for the unlimited assistance of Siyadebirena Wayu district Health Office for their cooperation and provision of the necessary information for this project. Last but not least, our acknowledgment goes to study participants, data collectors, and supervisors for their willingness to exert their efforts and time sacrificed for this study.

## Author Contributions

**Conceptualization:** Behailu Tariku Derseh, Natnael Mulushewa Tafese.

**Data curation:** Behailu Tariku Derseh, Natnael Mulushewa Tafese, Hazaratali Panari, Awraris Hailu Bilchut, Abel Fekadu Dadi.

**Formal analysis:** Behailu Tariku Derseh, Natnael Mulushewa Tafese, Hazaratali Panari, Awraris Hailu Bilchut, Abel Fekadu Dadi.

**Funding acquisition:** Natnael Mulushewa Tafese.

**Investigation:** Behailu Tariku Derseh, Natnael Mulushewa Tafese, Awraris Hailu Bilchut.

**Methodology:** Behailu Tariku Derseh, Natnael Mulushewa Tafese, Hazaratali Panari, Awraris Hailu Bilchut, Abel Fekadu Dadi.

**Project administration:** Behailu Tariku Derseh, Natnael Mulushewa Tafese.

**Resources:** Natnael Mulushewa Tafese.

**Software:** Behailu Tariku Derseh, Natnael Mulushewa Tafese, Hazaratali Panari, Awraris Hailu Bilchut, Abel Fekadu Dadi.

**Supervision:** Behailu Tariku Derseh.

**Validation:** Behailu Tariku Derseh, Natnael Mulushewa Tafese, Hazaratali Panari, Awraris Hailu Bilchut, Abel Fekadu Dadi.

**Visualization:** Behailu Tariku Derseh, Natnael Mulushewa Tafese, Awraris Hailu Bilchut, Abel Fekadu Dadi.

**Writing – original draft:** Behailu Tariku Derseh, Hazaratali Panari, Awraris Hailu Bilchut, Abel Fekadu Dadi.

**Writing – review & editing:** Behailu Tariku Derseh, Natnael Mulushewa Tafese, Hazaratali Panari, Awraris Hailu Bilchut, Abel Fekadu Dadi.

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
