## [Decision Letter · Decision Letter 0]

21 Jun 2021

PONE-D-20-36640

Behavioral and Environmental Determinants of Acute Diarrhea among Under-five Children from Public Health Facilities of Siyadebirena Wayu District, North Shoa Zone, Amhara Regional State, Ethiopia: Unmatched Case-control Study

PLOS ONE

Dear Dr. Derseh,

Thank you for submitting your manuscript to PLOS ONE. After careful consideration, we feel that it has merit but does not fully meet PLOS ONE’s publication criteria as it currently stands. Therefore, we invite you to submit a revised version of the manuscript that addresses the points raised during the review process.

The manuscript has been evaluated by two reviewers, and their comments are available below.

The reviewers have raised a number of concerns regarding the manuscript’s clarity and organization. They specifically request changes to improve the flow and coherence of the manuscript, paying particular attention to editing and grammatical issues. They also suggest improving the organization of the introduction, as well as including a better discussion of the study’s limitations and potential policy applications in the discussion. Please also note that updated references may be required.

Could you please carefully revise the manuscript to address all comments raised?

We look forward to receiving your revised manuscript.

Kind regards,

Avanti Dey, PhD

Staff Editor

PLOS ONE

Journal Requirements:

3. You indicated that you had ethical approval for your study. In your Methods section, please ensure you have also stated whether you obtained consent from parents or guardians of the minors included in the study or whether the research ethics committee or IRB specifically waived the need for their consent.

5. Thank you for stating the following financial disclosure: "No"

6. Thank you for stating the following in your Competing Interests section:  "No"

7. Please ensure that you refer to Figures 1 and 2 in your text as, if accepted, production will need this reference to link the reader to the figure.

8. Please upload a copy of Figure 3, to which you refer in your text on line 163. If the figure is no longer to be included as part of the submission please remove all reference to it within the text.

9. We note you have included a table to which you do not refer in the text of your manuscript. Please ensure that you refer to Table 1 in your text; if accepted, production will need this reference to link the reader to the Table.

10. Your ethics statement should only appear in the Methods section of your manuscript. If your ethics statement is written in any section besides the Methods, please move it to the Methods section and delete it from any other section. Please ensure that your ethics statement is included in your manuscript, as the ethics statement entered into the online submission form will not be published alongside your manuscript.

Reviewers' comments:

Reviewer's Responses to Questions

**Comments to the Author**

1. Is the manuscript technically sound, and do the data support the conclusions?

Reviewer #1: Yes

Reviewer #2: Partly

2. Has the statistical analysis been performed appropriately and rigorously? 

Reviewer #1: Yes

Reviewer #2: No

3. Have the authors made all data underlying the findings in their manuscript fully available?

Reviewer #1: No

Reviewer #2: No

4. Is the manuscript presented in an intelligible fashion and written in standard English?

Reviewer #1: No

Reviewer #2: Yes

5. Review Comments to the Author

Reviewer #1: The paper called “Behavioral and Environmental Determinants of Acute Diarrhea among Under-five Children from Public Health Facilities of Siyadebirena Wayu District, North Shoa Zone, Amhara Regional State, Ethiopia: Unmatched Case-control Study” by Behailu Tariku Derseh et al., is a multicentric unmatched case-control study. The study aimed to address the correlation of behavioral, socio-demographic, and sanitary variants with the diarrheic diseases in children between theirs 6-59 months old that visited a Daneba’s Health Center and the Daneba’s Hospital. The authors applied a questionnaire to 315 caregivers (105 were cases and 210 controls, finally 309 of them were included). The authors found that: low family income, hand washing without soap, non-treated drinking water consumption, poor caregiver’s knowledge of risk factors, disposal of fecal matter outside the latrine, and consumption of non-fresh food, had a statistically significant association with the development of acute diarrhea. The text of the manuscript requires a large improvement in the style

1) In the summary (and in other sections throughout the manuscript) the authors use the word “faces” instead of “feces”

2) Lines 40, in the summary the authors use two ways to express the odds ratio’s CIs

3) The introduction section does not have a fluid narrative. The text should begin with a global status explanation of the diarrheal diseases (mortality and morbidity), followed by regional or developing countries and the Ethiopian situation. The text has redundant information (not only in the introduction). For example, the definition of diarrhea and its consequences are addressed in lines 58-60, 63-67, 71-75, 142-144; the risk factors allude in lines 60-62, 69-71, 81-83, 92-94; and the access to treatment in lines 86-87, 95-96.

4) Line 78. Very confusing sentence “….acute diarrheal diseases secondary to acute gastroenteritis”

5) There other factors which have been associated with acute diarrhea in Ethiopia, the paper should include these factors (with the respective references) in the introduction or discussion sections.

6) Lines 76-77. Rewrite the sentence

7) Lines 92-94: Rewrite the sentence

8) Line 105: It is unclear why the authors describe the number of males in an Ethiopian region.

9) The aim of the study is in the abstract but not in the introduction.

10) Lines: 39, 106, 314-316, Some terms are local for the Ethiopian readers, however for an international audience they are confusing. For example, the ETB currency can be explained in poverty threshold in Euros or US dollars. The term “urban Kebele” should be explained.

11) Figure 1 is low quality

12) In lines 87-91 (also 96-99), the authors mention some Ethiopian studies about diarrhea prevalence. The authors must discuss in more detail the differences between settings, incomes, study design, the accomplishment of WHO recommendations, and other variables that explain such variances.

13) Line 134, replace “was” with “were”

14) Line 144, explain that controls came from children attending health centers

15) Line 197, it will be useful to have a supplemental document showing the questionnaire, it is important to see the questions that the authors used to assess the parents’ knowledge about risk factors

16) The authors need to explain the "institution-based" design. (line 118)

17) Legend of Table 3 should be rewritten

18) Tables should be formatted following the guidelines of the journal and legends should be self-explanatory. The acronyms such as COR and AOR should be explained in the legend.

19) In table 5, statistically significant values seem to be written in bold. This should be indicated in the legend.

20) The methodology section has redundancies in the text. First, the case definition is in lines 122-124, 131-132, 142-144 with slight differences that confused the readers. The same case is the definition of controls (lines 126-127, 144-145). Third, the dates are mentioned in lines 118-119, and 131. The estimation of the sample size is segmented within the text in lines 135-141 and 150-157. As mentioned, this segmentation of the information and redundancy confuse the readers.

21) The contribution of the authors should be in the respective section, please check lines 205 and 219.

22) The staff training is mentioned twice in 201-203 lines and 212-14 lines.

23) The text needs to be more concise, please synthesize the methods section and avoid redundancies. (lines 197-242)

24) The authors calculated a sample size of 315 participants however, they include only 309 (line 253). Explain the reasons for withdrawing the rest of the participants in the result section (it is slightly mentioned in lines 145-148).

25) The results section has several redundancies between the text and the tables. The tables need to be interpreted for the authors to highlight the important findings. As an example, the monthly income is mentioned in table 2, and in table 5 with the same data. Table 5 format is an improvement of Tables 2, 3, and 4 and permits the reader to get more information and engagement. The authors should use the format of Table 5 for all the tables because it includes the statistical analysis with all the other variables.

26) The discussion section should be rewritten to synthesize some ideas. The authors of this study use the statistical significance data (AOR) of other similar studies to compare their findings.

27) The study findings are suitable to propose recommendations to reduce the incidence of diarrheal diseases in this Ethiopian setting. The discussion needs a statement for recommendations, to gather all the ones disperse within the text (lines 337-339, 361-366, 376-379, 392-393, 421-430). The authors do not address the importance of public health policies and actions (and maybe the most supported) to improve sanitation and hygiene.

28) One of the main limitations of this study is that it does not consider causative agents and the pathogens involved in different factors may be different. For example, handwashing (331-339), drinking water contamination (347-349), no fresh food consumption (370-379), among others. This limitation has to be mentioned in the text. This approach can guide the recommendations for other studies.

29) Lines 380-401 should be massively condensed

30) Line 410 “Not” should be “not”

Reviewer #2: Date 19/5/2021

Comments for authors or editors:

Thank you for asking to review your paper titled “Behavioral and Environmental Determinants of Acute Diarrhea among Under-five Children from Public Health Facilities of Siyadebirena Wayu District, North Shoa Zone, Amhara Regional State, Ethiopia: Unmatched Case-control Study.”

General critique

To me, the paper is important to enhance and to recall the existing body of knowledge about behavioral and environmental determinants of acute diarrhea among Under-five Children. However, I have several concerns that need to be address before I can say that it is ready for publication. Besides, I will also suggest some editorial changes.

Abstract

The abstract is relatively good.

Page 2 line 33 “Methods: “Facility-based unmatched case-control study design was employed from March 12, 34 2019, to May 12, 2019.”

Better to be “Facility-based unmatched case-control study was conducted from March 12, 2019, to May 12, 2019.”

Page 2 line 37 “The binary logistic regression model was employed to evaluate the independent effect of the predictor variables on acute childhood diarrhea”.

Better to be “To analysis the data, binary logistic regression and multivariable logistic regression analysis was conducted.”

Introduction

With the exception of some repeated and an attractive sentences, the introduction part of the paper is sound. Moreover, this part lacks essential references, particularly for those sentences that states figure like in page (P) 3, line (L) 76-77 which states “Annually, it is estimated that 1.3 million deaths are associated with diarrheal diseases with the most occurring in resource-limited countries” and P3, L 67-68 which states “Acute diarrhea is a major public health problem in the world. Next to pneumonia, it is the leading cause of death in children under five years old.” And soon.

P3,L 58-60 which states “It is caused by bacterial, viral, and parasitic organisms and is usually a symptom of gastrointestinal infection which can be caused by a variety of bacterial, viral, and parasitic organisms. Please change it into” It is caused by bacterial, viral, and parasitic organisms and is usually causes gastrointestinal infection.”

P3 L62, please add the word “practice” after the word “sanitation”

P3 L82, delete the word “the”

P4 L88 add (,) after the word” revealed that”

Methods

The method part of the paper is relatively sequential, logical and well-constructed except the analyzing method.

Study area:

Although it is not supported by references, the “topic of study area” provided good information to readers.

Data collection tools and methods

P8 L201, “Data was collected by a nurse who works at the under-five OPD/inpatient pediatrics ward of the health center and hospital.” Do you mean the data was collected by 1 nurse? Did you provided training for 1 data collector? I do not think so. See it.

P8 L208, delete the word “the” and replace it with “A”

Data Processing and Analysis

Here I have a great concerned.

P8 L235-238, stated that “To avoid an excessive number of variables and unstable estimates in the subsequent model, only variables with a p-value < 0.20 were kept in the analyses. Independent variables which result in a p-value less than 0.20[15] in an unadjusted model are candidates to be considered for the final multivariable model.” I am not satisfying with the reason that the authors provided. Rather, in my opinion, this reason caused to reject may core, known and scientifically assured determinant variables that affect positively or negatively to diarrheal disease in the analysis like Rotavirus vaccine, type of sources of drinking-water etc..

Scientifically, in binary logistic regression analysis, a variable which has a P-value < 0.05 must be included in the multivariable analysis. And based on the result of the multivariable analysis any one can discuss and conclude his/her findings. However, the authors could not follow this idea.

Results

This part has many grammatical problems and figure which are not present in the respected tables. In short, it is written in a poor manner.

P10 L252 delete (’) which is located after the word “respondents.”

P10 L255 states “ Forty-nine (47.6 %)” which is wrong.

P10 L 256 – 257, stated “Regarding the age of children, the average was 32.31 257 (±5.6) years for cases and 32.18 (±5.6) for controls. Can we call children whose age 32.3 and 32. 18 years? See it.

P12 L267, stated “Out of the respondents….”. Please change in to “Out of the total respondents….”

P12 L269 put (.) after the word “water” and start with But….

However, “But 87(84.5%) of cases and 270 43(20.9%) of controls, did not treat drinking water at home.” Those figures are not present in the table. Why?

P13 L280, stated “poor hand washing practices”. What are your measurements and cut-off points to say poor or good? The same to that of “Knowledge of major risk factors” (Table 4)

P13 L 284-285, stated “… and 41(39.8%) of cases and 33(16%) 285 of controls did not receive vitamin-A supplementation”. The figures are not present in the table 4. Why?

Discussion

The discussion is very poor. It is based on results comparison. The discussion must include practical implications and “why” these results are relevant.

P18 L364, make the 2 references into 1.

P18 L398 stated “Therefore, awareness creation among mothers/ caretakers about the importance of action taken to halt acute diarrhea should be given attention.” To me, this is recommendation. Take it to its topic.

P18-19 L396- 398, states “Similarly, the odds of developing diarrheal morbidity were higher among children whose mothers/caretakers had better knowledge about the causes of diarrhea and had hand washing practice (AOR = 2.46, 95% CI: 1.07, 5.63) [23].” How it could be similar? This result is also contrary to facts.

Conclusion

The conclusion is good. However, it might be better than this.

P19 L426-427, stated “Providing simple and “easy to understand” information to the mothers/caretakers on major risk factors of acute diarrhea.” Unclear, modify this phrase.

Availability of data

The authors stated “The datasets used and/or analyzed during the current study are available in this manuscript. However, the corresponding author on reasonable request can submit the original dataset.”

Why not submitted without request during the submission time?

References

Some of the references used were too old. Moreover, see the reference style of the journal because some of these are not according the journal.

6. PLOS authors have the option to publish the peer review history of their article (what does this mean?). If published, this will include your full peer review and any attached files.

Reviewer #1: **Yes: **G. Trueba

Reviewer #2: **Yes: **Dr. Aderajew Mekonnen Girmay

---

## [Author Response · Author response to Decision Letter 0]

14 Jul 2021

Response to the editor/reviewers

PONE-D-20-36640

Behavioral and Environmental Determinants of Acute Diarrhea among Under-five Children from Public Health Facilities of Siyadebirena Wayu District, North Shoa Zone, Amhara Regional State, Ethiopia: Unmatched Case-control Study

Dear Dr. Avanti Dey, Dr. G. Trueba, and Dr. Aderajew M, thank you very much for your comprehensive suggestions and comments. We are lucky to have you as an editor and reviewer. We have gone through each and every concern you have provided. Kindly addressed all the points raised and presented in the table below. Moreover, after taking all corrections based on reviewers (2) comments, we considered 1 – 10 editor(s) concerns too. Therefore, (1) we followed PLOS ONE style requirements, (2) we included supplementary information (e.g., S1 annex), (3) consents considered in method section, (4) data availability statement-revised (5) Funding statement-removed from the manuscript (6) Competing interests-removed from the manuscript & included in online system (7) Figures 1 and 2 properly referenced in the text, and the map was removed (8) Figure 3 was deleted since it's editorial error, (9) Table 1 referred, and (10) Ethics were included in method section. Finally, we attached 3 files, namely; response to the editor, manuscript with track change, the revised version of the manuscript including the supporting materials (4).

---

## [Editor Report · Decision Letter 1]

17 Sep 2021

PONE-D-20-36640R1

Behavioral and Environmental Determinants of Acute Diarrhea among Under-five Children from Public Health Facilities of Siyadebirena Wayu District, North Shoa Zone, Amhara Regional State, Ethiopia: Unmatched Case-control Study

PLOS ONE

Dear Dr. Derseh,

Thank you for submitting your manuscript to PLOS ONE. After careful consideration, we feel that it has merit but does not fully meet PLOS ONE’s publication criteria as it currently stands. Therefore, we invite you to submit a revised version of the manuscript that addresses the points raised during the review process.

We look forward to receiving your revised manuscript.

Kind regards,

Gabriel Trueba, PhD

Academic Editor

PLOS ONE

Journal Requirements:

Additional Editor Comments (if provided):

Page 2 lines 42 and 46 Replace the sentence "consuming a child with leftover food" wirh "children consuming left-over food stored at room temperature"
---

## [Author Response · Author response to Decision Letter 1]

26 Oct 2021

We included response to the editor file to address the points raised during review. Thank you.

---

## [Editor Report · Decision Letter 2]

28 Oct 2021

Behavioral and environmental determinants of acute diarrhea among under-five children from public health facilities of Siyadebirena Wayu district, north Shoa zone, Amhara regional state, Ethiopia: unmatched case-control study

PONE-D-20-36640R2

Dear Dr. Derseh,

We’re pleased to inform you that your manuscript has been judged scientifically suitable for publication and will be formally accepted for publication once it meets all outstanding technical requirements.

Kind regards,

Gabriel Trueba, PhD

Guest Editor

PLOS ONE
---

## [Editor Report · Acceptance letter]

12 Nov 2021

PONE-D-20-36640R2 

Behavioral and environmental determinants of acute diarrhea among under-five children from public health facilities of Siyadebirena Wayu district, north Shoa zone, Amhara regional state, Ethiopia: unmatched case-control study 

Dear Dr. Derseh:

I'm pleased to inform you that your manuscript has been deemed suitable for publication in PLOS ONE. Congratulations! Your manuscript is now with our production department. 

Kind regards, 

on behalf of

Dr. Gabriel Trueba 

Guest Editor

PLOS ONE